# Study of Antimicrobial Resistance, Biofilm Formation, and Motility of *Pseudomonas aeruginosa* Derived from Urine Samples

**DOI:** 10.3390/microorganisms11051345

**Published:** 2023-05-19

**Authors:** Telma de Sousa, Michel Hébraud, Olimpia Alves, Eliana Costa, Luís Maltez, José Eduardo Pereira, Ângela Martins, Gilberto Igrejas, Patricia Poeta

**Affiliations:** 1Department of Genetics and Biotechnology, University of Trás-os-Montes and Alto Douro (UTAD), 5000-801 Vila Real, Portugal; telmas@utad.pt (T.d.S.);; 2Microbiology and Antibiotic Resistance Team (MicroART), Department of Veterinary Sciences, University of Trás-os-Montes and Alto Douro (UTAD), 5000-801 Vila Real, Portugal; 3Functional Genomics and Proteomics Unit, University of Trás-os-Montes and Alto Douro (UTAD), 5000-801 Vila Real, Portugal; 4Associate Laboratory for Green Chemistry (LAQV), Chemistry Department, Faculty of Science and Technology, University Nova of Lisbon, 2829-516 Lisbon, Portugal; 5Université Clermont Auvergne, INRAE, UMR Microbiologie Environnement Digestif Santé (MEDiS), 60122 Saint-Genès-Champanelle, France; 6Medical Centre of Trás-os-Montes and Alto Douro, Clinical Pathology Department, 5000-801 Vila Real, Portugal; 7Veterinary and Animal Research Centre (CECAV), University of Trás-os-Montes and Alto Douro (UTAD), 5000-801 Vila Real, Portugal; 8Department of Veterinary Sciences, University of Trás-os-Montes and Alto Douro (UTAD), 5000-801 Vila Real, Portugal; 9Department of Zootechnics, University of Trás-os-Montes and Alto Douro (UTAD), 5000-801 Vila Real, Portugal; 10Associate Laboratory for Animal and Veterinary Sciences (AL4AnimalS), University of Trás-os-Montes and Alto Douro (UTAD), 5000-801 Vila Real, Portugal

**Keywords:** *Pseudomonas aeruginosa*, biofilm, motility, antibiotic resistance, virulence

## Abstract

*Pseudomonas aeruginosa* causes urinary tract infections associated with catheters by forming biofilms on the surface of indwelling catheters. Therefore, controlling the spread of the bacteria is crucial to preventing its transmission in hospitals and the environment. Thus, our objective was to determine the antibiotic susceptibility profiles of twenty-five *P. aeruginosa* isolates from UTIs at the Medical Center of Trás-os-Montes and Alto Douro (CHTMAD). Biofilm formation and motility are also virulence factors studied in this work. Out of the twenty-five *P. aeruginosa* isolates, 16% exhibited multidrug resistance, being resistant to at least three classes of antibiotics. However, the isolates showed a high prevalence of susceptibility to amikacin and tobramycin. Resistance to carbapenem antibiotics, essential for treating infections when other antibiotics fail, was low in this study, Notably, 92% of the isolates demonstrated intermediate sensitivity to ciprofloxacin, raising concerns about its efficacy in controlling the disease. Genotypic analysis revealed the presence of various β-lactamase genes, with class B metallo-β-lactamases (MBLs) being the most common. The *bla*_NDM_, *bla_S_*_PM_, and *bla*_VIM-VIM2_ genes were detected in 16%, 60%, and 12% of the strains, respectively. The presence of these genes highlights the emerging threat of MBL-mediated resistance. Additionally, virulence gene analysis showed varying prevalence rates among the strains. The *exo*U gene, associated with cytotoxicity, was found in only one isolate, while other genes such as *exo*S, *exo*A, *exo*Y, and *exo*T had a high prevalence. The *tox*A and *las*B genes were present in all isolates, whereas the *las*A gene was absent. The presence of various virulence genes suggests the potential of these strains to cause severe infections. This pathogen demonstrated proficiency in producing biofilms, as 92% of the isolates were found to be capable of doing so. Currently, antibiotic resistance is one of the most serious public health problems, as options become inadequate with the continued emergence and spread of multidrug-resistant strains, combined with the high rate of biofilm production and the ease of dissemination. In conclusion, this study provides insights into the antibiotic resistance and virulence profiles of *P. aeruginosa* strains isolated from human urine infections, highlighting the need for continued surveillance and appropriate therapeutic approaches.

## 1. Introduction

*Pseudomonas aeruginosa* is one of the most opportunistic and common Gram-negative bacteria, being a main source of human, animal, and plant infections [1]. *P. aeruginosa* has a great diversity of pathoadaptive characteristics and virulence mechanisms that promote its colonization, survival, and proliferation in different environments (human, animal, and plant infections) [2,3]. The first step of the infectious process is the binding of *P. aeruginosa* to the epithelial host cell. The machinery responsible for bacterial motility, flagella and pili, act as tethers in the initial attachment of bacteria to surface glycolipids of epithelial cells [4]. Upon contact with the host cell, the type III secretion system (T3SS) is activated, allowing *P. aeruginosa* to inject secreted toxins (ExoS, ExoY, ExoT, and ExoU) through a syringe-like device directly into the host’s cytoplasm. These effector proteins participate, to varying degrees, in cytotoxicity during the invasion and dissemination of this bacterium [5,6]. Other systems are also activated during the infection process, such as type II secretion system (T2SS), which secrete other virulence factors such as elastase, exotoxin A, alkaline phosphatase, and phospholipase C into the extracellular space. These virulence factors aid in the invasion process, destroying the protective glycocalyx of the respiratory epithelium and exposing epithelial ligands to *P. aeruginosa* [7,8]. The *tss*C gene in *P. aeruginosa* is an important component of the type VI secretion system (T6SS), a complex nanomachine that allows bacteria to deliver toxins or other effector molecules into target cells [9]. The TssC protein is a structural component of the T6SS apparatus, and it plays a critical role in the assembly and function of the system. The TssC protein is required for the stability of the T6SS apparatus, and it helps anchor the baseplate complex to the bacterial cell envelope. In addition, it is necessary for the formation of the Hcp tube, which is a key component of the T6SS that delivers effector molecules to target cells. The TssC protein is also important for the virulence of *P. aeruginosa*, as it allows the bacterium to deliver toxins or other virulence factors into host cells. Mutants lacking the TssC gene have been shown to be attenuated in virulence both in vitro and in vivo [9,10].

In *P. aeruginosa*, both in acute and chronic infections, there is an infinity of regulatory systems at its disposal that allow it to adapt to its environment and defend itself against the host. Quorum sensing (QS) is an example of such a system and demonstrates the adaptability of *P. aeruginosa* [11]. These systems allow *P. aeruginosa* isolates to grow in biofilms, communicating and coordinating the production and release of virulence factors [12]. The *P. aeruginosa* genome contains a large number of genes that encode outer membrane proteins involved in adhesion, motility, antimicrobial efflux, virulence factor export, and environmental detection by two-component systems. Thus, it is not surprising that the genome of this bacterium contains a variety of predicted genes that are involved in the regulation of virulence genes [13].

The ability of *P. aeruginosa* to move by swimming, swarming, and twitching is crucial for the bacterium to colonize both living and non-living surfaces, as well as for its spread across these surfaces [6]. Many planktonic organisms can initially colonize a surface using a flagellum to swim towards the surface and attach themselves via bacterial adhesins, such as type IV pili and flagella. In the case of *P. aeruginosa*, it can undergo the flagellum-mediated swimming motility and the surface-associated swarming and twitching motilities, which are predominantly mediated by hyperflagellation and type IV pili, respectively [14,15]. Pili are hair-like structures on the surface of bacteria that are important for attachment to surfaces and for motility. *P. aeruginosa* has two types of pili: type IV pili (T4P) and type IVa pili (T4aP) [16,17]. The genes *pil*A and *pil*B are involved in the biosynthesis and assembly of T4P in *P. aeruginosa*. PilA is the major subunit protein of T4P, and it is responsible for the structure and function of the pili. The *pil*A gene is located in the pilin gene cluster of *P. aeruginosa*, which also contains several other genes involved in the biosynthesis and assembly of T4P. The expression of *pil*A is regulated by several factors, including quorum sensing and environmental signals [17,18]. PilB is an ATPase that is involved in the assembly and disassembly of T4P. The pilB gene is located outside of the pilin gene cluster, and its expression is also regulated by quorum sensing and environmental signals [17,19]. Motility also contributes to the formation of biofilms, which are structured communities of bacteria associated with the surface [15]. These bacterial communities are highly persistent and protected from adverse conditions owing to their specialized structures. Normally, biofilms are enclosed within the extracellular polymer matrix, leading to the production and specific secretion of exopolysaccharides, DNA, and/or protein by the bacterium itself [1]. Biofilm formation is the source of a major problem for the host, as these biofilms are often resistant to antibiotics, phagocytosis, and surfactants, making them difficult to remove once established [20]. The biofilm formation process in *P. aeruginosa* is regulated by a number of genetic and environmental factors. One of the key systems involved in biofilm formation is the quorum sensing system. This system allows bacteria to communicate with one another and coordinate their behavior based on cell density [21,22]. In *P. aeruginosa*, the quorum sensing system is mediated by a number of small signaling molecules, including acyl-homoserine lactones (AHLs) and 2-heptyl-3-hydroxy-4-quinolone (PQS). Another important factor in biofilm formation is the production of extracellular polymeric substances (EPS). These substances are secreted by bacteria and form a matrix that helps to anchor the bacteria to surfaces and protect them from environmental stresses, including alginate, Pel, and Psl [22]. The genome of *P. aeruginosa* contains a number of features that are responsible for biofilm formation. For example, the genes encoding the quorum sensing system and EPS production are located in specific regions of the genome called “pathogenicity islands”. In addition, *P. aeruginosa* has a large and complex genome that contains numerous genes involved in regulating biofilm formation and responding to environmental cues. Overall, the biofilm formation process in *P. aeruginosa* is a complex and multifactorial process that involves a range of genetic and environmental factors [6,23].

The prevalence of multidrug-resistant strains in *P. aeruginosa* is a cause for concern, as this bacterial species has developed a significant number of such strains. The increasing multidrug resistance of *P. aeruginosa* represents a major threat to public health and well-being. Modification enzymes can modify antibiotics and render them ineffective. For example, this bacterium produces aminoglycoside-modifying enzymes, such as AAC(6’)-Ib and AAC(3)-I, which can acetylate or adenylate aminoglycoside antibiotics, respectively, and reduce their efficacy [24,25]. The resistance to carbapenems in *P. aeruginosa* is often associated with the acquisition of specific genes that produce carbapenemases, enzymes that can break down the carbapenem antibiotics. These genes are often located on mobile genetic elements such as plasmids, transposons, or integrons, which can facilitate their spread between different bacterial strains and species. The most common carbapenemases found in *P. aeruginosa* are metallo-beta-lactamases (MBLs), including the VIM, IMP, and NDM types, and serine carbapenemases, including the OXA-type enzyme. These carbapenemases are often encoded by genes located on mobile genetic elements and can confer resistance to other antibiotics as well, making the treatment of infections caused by these strains challenging [26]. This emerging risk hampers the use of traditional antibiotics and increases the effectiveness of antimicrobial treatments [27]. The study of the pathogenicity of *P. aeruginosa* becomes crucial for the identification of compounds that prevent bacterial motility and thus combat the potential for colonization of different surfaces [6,15]. Analyzing urine samples provides specific information on the presence and behavior of *P. aeruginosa* in urinary tract infections. *P. aeruginosa* is responsible for 7–10% of urinary tract infections (UTIs) in hospitalized patients, making UTIs one of the most common ailments among patients receiving medical treatment [28,29]. In fact, UTIs account for between 20 and 49% of all nosocomial infections [30,31]. The pathogenesis of urinary tract infections caused by *P. aeruginosa* is also influenced by virulence factors produced by the bacterium. *P. aeruginosa* produces a variety of extracellular products, such as pyocyanin, exotoxin A, and elastase, which can damage the host tissue and impair the immune system response. Additionally, *P. aeruginosa* can produce flagella and type IV pili, which allow it to move through the urinary tract and colonize the bladder or kidney [32,33].

Another problem associated with urinary tract infections is catheter-associated urinary tract infections (CAUTIs). When a urinary catheter is inserted, it provides a surface for *P. aeruginosa a* to colonize and grow, leading to CAUTIs. *P. aeruginosa* can adhere to the surface of a urinary catheter and produce a biofilm. The biofilm formation by *P. aeruginosa* can facilitate its survival in the urinary tract by protecting it from the host immune system and antimicrobial agents. Furthermore, biofilms can act as a reservoir for antibiotic resistance genes, leading to the development of multidrug-resistant *P. aeruginosa* strains [34,35].

Understanding the prevalence and mechanisms of antibiotic resistance in infections caused by *P. aeruginosa* is important for guided treatment decisions and for reducing the spread of resistant strains [36]. It is important to note that studying other types of specimens, such as blood or tissues, could provide different information and may be appropriate in other relevant research questions. In this study, we aimed to focus on the *P. aeruginosa* isolated and derived from urine samples in the Medical Centre of Trás-os-Montes and Alto Douro and to characterize its antimicrobial resistance, biofilm formation, and motility.

## 2. Materials and Methods

### 2.1. Samples and Bacterial Isolates

Twenty-five *Pseudomonas aeruginosa* samples were isolated from urinary tract infection in the Medical Centre of Trás-os-Montes and Alto Douro (CHTMAD), Vila Real, Portugal, between January 2022 and March 2022. All strains were isolated using VITEK 2^®^ COMPACT (bioMérieux), and its identification was confirmed in the medical microbiology laboratory by seeding on *Pseudomonas* Agar Base supplemented with CN (Liofilchem, Rosetodegli, Abruzzi, Italy) medium at 37 °C for 24–48 h. The isolates were cryopreserved at −20 °C in skim milk.

### 2.2. Biofilm Formation and Biomass Quantification

The bacterial adhesion of all isolates was assessed using a microtiter plate-based assay as previously described with some modifications [37]. To conduct the experiment, one colony from each bacterial culture grown overnight on brain heart infusion (BHI) agar was suspended in Luria-Bertani (LB) broth and incubated at 37 °C for 24 h. Then, the bacterial suspension was diluted to 0.5 on the McFarland scale using Tryptic Soy Broth (TSB). An amount equaling 100 µL of each bacterial suspension, along with a negative control of sterile TSB and a positive control of *Pseudomonas aeruginosa* ATCC 27853^®^, were inoculated into eight wells of a flat-bottom polystyrene microtiter plate, with each well serving as a technical replicate. The plate was then incubated at 37 °C for 24 h. After incubation, the plate was washed twice with distilled water and allowed to dry at room temperature. Then, 100 µL of 0.1% (*v*/*v*) crystal violet (CV) was added to each well and incubated for 10–15 min. The CV was then washed out of the plate, and the wells were allowed to dry. For qualitative assays, photographs of the wells were taken when dry. To quantify the biofilm biomass, 100 µL of 30% (*v*/*v*) acetic acid was added to each well to solubilize the CV. The optical density was then read at 630 nm using a blank of uninoculated 30% acetic acid and a microplate reader (BioTek ELx808U). Biofilm-formation ability was considered positive at a cut-off level of 0.078, which was determined arbitrarily as the mean for the negative control (culture medium, 0.058) plus three standard deviations (0.006).

The levels of biofilm production were classified as follows: weak biofilm formers had an optical density between 0.078 and 0.157 (2 times the negative control), moderate biofilm formers had an optical density between 0.157 and 0.314 (4 times the negative control), and strong biofilm formers had an optical density greater than 0.314.

### 2.3. Motility Assays

#### 2.3.1. Swimming Motility

The isolates were grown on LB agar at a temperature of 37 °C for 24 h, after which a single colony was selected and subcultured on LB broth. The swimming medium was then prepared by adding 0.3% (*w*/*v*) agar to the LB broth [15]. This mixture was used to inoculate the swimming plates with a five µL sample of the bacterial broth culture containing approximately 10^8^ CFU/mL. To assess motility within the semi-solid agar, the inoculum was placed directly in the center of the agar on the swim plates. [38].

#### 2.3.2. Swarming Motility

Isolates were grown on LB agar at 37 °C for 24 h prior to preparation of a single colony subculture into LB broth. Swarm media was prepared and inoculated as follows: nutrient broth (Oxoid Australia Pty. Ltd., Adelaide, Australia) supplemented with 0.5% agar (*w*/*v*) and 0.5% D-Glucose (*w*/*v*) (Sigma-Aldrich Pty. Ltd., Castle Hill, Australia) [15]. For swarm, plates were inoculated with five µL of bacterial broth culture representing approximately 10^8^ CFU/mL. The inoculum was placed on the agar surface (center) enabling visualization of motility across the agar surface [39].

#### 2.3.3. Twitching Motility

Isolates were grown on LB agar at 37 °C for 24 h prior to preparation of a single colony subculture into LB broth. Twitch media was prepared and inoculated as follows: LB broth supplemented with 1.0% (*w*/*v*) agar [15]. For twitching motility, plates were inoculated with five µL of bacterial broth culture representing approximately 10^8^ CFU/mL. Five µL was inoculated deep into the agar with a micropipette so that the tip touched the agar-plate interface, and motility at this interface was subsequently measured [40].

### 2.4. Interpretation of Motility Assays

All the motility assays were performed in triplicate. The diameter in mm of the growth zone was measured after 24 h of incubation. From these individual measurements, the average area for each representative isolate was calculated to determine motility characteristics. Of note, the growth area is required to be at least 10% of the control PAO1. Another criterion used was the total growth area, which must be ≥20 mm^2^ to be considered mobile. If the value was less than 20 mm^2^, we concluded that, although the bacteria were able to grow, they were not mobile [41,42]. *Pseudomonas aeruginosa* ATCC^®^ 27,853 was included in all plates as a positive control and *Klebsiella pneumoniae* as a negative control.

### 2.5. Antimicrobial Susceptibility Testing

The phenotypic susceptibility profile characterization of the isolates was performed using the Kirby–Bauer disk diffusion method in concordance with EUCAST standards (2022). Twelve antibiotics disks were tested: ceftazidime (10 μg), cefepime (30 μg), amikacin (30 μg), gentamicin (10 μg), tobramycin (10 μg), doripenem (30 μg), imipenem (10 μg), aztreonam (30 μg), ciprofloxacin (50 μg), piperacillin (30 μg), and ticarcillin-clavulanic acid (85 μg). For colistin (4 mg/L) and piperacillin–tazobactam (16 mg/L), the minimum inhibitory concentration (MIC) method was performed by microdilution broth following EUCAST norms. For colistin, EUCAST advises using the microdilution method instead of the disk. Due to the relevance of piperacillin–tazobactam, microdilution was also performed.

### 2.6. DNA Extraction

The method used was the boil method [43]. Briefly, two to three colonies of overnight growth bacteria were used. The colonies were put in a test tube containing 500 µL of distilled water and boiled for 8 min in a water bath. After being vigorously vortexed, the samples were centrifuged for 2 min at 12,000 rpm and the pellet discarded. Total DNA concentration was determined using a NanoDrop system. A measurement of absorbance was taken at wavelengths of 260 and 280 nm to determine the concentration of nucleic acids. Each unit of optical density is equal to 50 μg/mL of double-stranded DNA. The PCR was conducted using a concentration of 200 μg/mL for each sample. All DNA samples had the optimal purity of 1.8 and 2.

### 2.7. Antimicrobial Resistance and Virulence Genes

All isolates were selected for the presence of antimicrobial resistance genes according to the result obtained by phenotypic resistance. All bacterial genomic DNA were used as templates for PCR amplification of the 16S rDNA gene for the confirmation of *P. aeruginosa.* The two primers used were 27F (5′ AGAGTTTGATCCTGGCTCAG-3ʹ) and 1495R (5′ CTACGGCTACCTTGTTACGA- 3ʹ) for forward primer and reverse primer, respectively [44]. According to the phenotypic resistance profile, each isolate was screened using PCR for the presence of the following antimicrobial resistance genes: *bla*_TEM_, *bla*_SHV_, *bla*_CTX_, *bla*_PER_, *bla*_SME_, *bla*_KPC_, *bla*_Smp_, *bla*_Vim_, *bla*_Vim-2_, *bla*_NDM_, *bla*_OXA_, *aac*(6′)-Ie-*aph*(2″)-Ia, *aph*(3′)-IIIa, acc3I, aac3II, aacIII, aac3IV, *ant*(4′)-Ia, and *ant*(2′)-Ia. All isolates were screened for genes encoding virulence factors by PCR: *pil*B, *pil*A, *apr*A, *tox*A, *tss*C, *plc*H, *las*A, *las*B, *las*R, *lasI*, *exo*U, *exo*S, *exo*A, *exo*Y, *exo*T, *rhl*R, *rhl*I, *rhl*A/B and *alg*D. All primer sequences are shown in Table 1. Several strains from the molecular genetics laboratory were used as positive controls for each gene, and Milli Q water as a negative control.

### 2.8. Statistical Analysis

For variables that showed considerable skewness, data were transformed into (log (1 + x)), and linear modelling on the log scale was performed. To analyze the reproducibility of the motility tests, Bland–Altman plots were used. For the analysis of the movements, ANOVA followed by Tukey’s was used to compare as averages. Statistical analyses were performed using the commercially available statistical packages; SPSS version 25 (IBM SPSS Statistics, Chicago, IL, USA) and one-way ANOVA followed by Dunnett’s multiple comparison test was performed using GraphPad Prism version 9.3.1 for Windows (GraphPad, San Diego, CA, USA).

## 3. Results and Discussion

In this study, *P. aeruginosa* strains were isolated from human urine infections. Among the twenty-five *P. aeruginosa* isolates, 16% were resistant to at least three classes of antibiotics at the phenotypic level (Table 2), which makes them, by definition, multidrug resistant. The isolates showed a high prevalence of antimicrobial susceptibility to amikacin (*n* = 25) and tobramycin (*n* = 25). In contrast, only a few isolates were resistant to piperacillin–tazobactam (*n* = 5), ceftazidime (*n* = 4), and gentamicin (*n* = 4). Most of the isolates showed intermediate sensitivities to most of the antibiotics tested. In our study, the prevalence of resistance to carbapenem antibiotics was low, which agrees with other reports in the literature [64,65,66]. However, other studies demonstrated a high prevalence of resistance to carbapenems in *P. aeruginosa* strains, such as 55.5% in *P. aeruginosa* isolates at a German university medical center, and percentages of resistance to imipenem and meropenem in a Spanish hospital were 88% and 82%, respectively [67,68]. The high resistance to carbapenem antibiotics is alarming at the public health level, since these antibiotics are essentials for the treatment of infections in which there are no more effective low-class antibiotics [68]. Our observations demonstrated both concordant and discordant findings when compared to existing reports in the scientific literature.

Of all the isolates, 92% demonstrated intermediate sensitivity to ciprofloxacin, while the remaining 8% exhibited resistance. The prevalence of intermediate resistance to ciprofloxacin, coupled with the absence of fully susceptible isolates, poses a challenge, as it may compromise the efficacy of this antibiotic, thereby limiting its ability to satisfactorily control the disease(s). In other studies, 40.5% [69], 50% [70], and 72.41% [71] susceptibility to ciprofloxacin were observed. Thus, the use of this antibiotic deserves additional attention and should be used only when necessary as an alternative therapeutic agent for isolates resistant to other antibiotics [64,71].

The genotypic results for the rDNA 16S gene were positive for all isolates, allowing us to conclude that all of them were *P. aeruginosa*, as expected.

Despite conducting a characterization of several genes, the isolates exhibited a certain degree of diversity. Nevertheless, isolates 7, 13, 14, and 19 demonstrated the presence of *bla*_KPC_ and *opr*D genes, while isolates 21–25 had the *bla*_SPM_ and *opr*D genes. Based on this, it can be inferred that these isolates are more similar to each other than to the remaining ones. This commonality suggests that these isolates may be more closely related to one another genetically than to the other isolates that have a different combination of genes. It is important to note, however, that this inference is based solely on the presence or absence of these specific genes and not on a more comprehensive genetic analysis. Further investigation would be necessary to confirm any relationships between the isolates.

All the strains encoded genes to produce β-lactamases from different classes. The class B MBLs β-lactamases were the most common in the strains due to the presence of the *bla*_NDM_, *bla*_SPM_, and bla_VIM-VIM2_ genes. According to a study carried out by Picão et al., early detection of MBL-producing genes may contribute to controlling the spread of multidrug-resistant isolates [72]. NDM is an enzyme capable of hydrolyzing a broad range of antibiotics, including carbapenems, penicillins, and cephalosporins [73]. NDM-1-producing *P. aeruginosa* is clonally disseminated from countries with a history of *bla*_NDM_ presence [50]. Thus, the spread of this gene to non-endemic countries through international travel and its transmission is alarming. In our study, the genes bla_NDM_, *bla*_SPM_, and bla_VIM-VIM2_ were detected in 16%, 60%, and 12%, respectively. Contrary to other works, the percentage of bla_VIM-VIM2_ is smaller; however, the difference can be explained by the existence of numerous variants of this gene [52]. Nonetheless, we should not disregard the low detection of this gene, as its spread can occur quickly, since it is one of the most prevalent emerging types of MBL genes in the world. Furthermore, the *bla*_SPM_ gene have already been reported in ICU patients with colonization or invasive disease [74]. The presence of the *bla*_KPC_ gene responsible for resistance to class A carbapenems was detected in 52% of the samples. A study carried out in Iran also indicated the presence of KPC-producing bacteria in a hospital environment, with a high incidence rate [75].

Only one isolate was resistant to aminoglycosides, namely gentamicin, which harbored the resistance genes *aph*(3′)-IIIa. With regard to colistin, although all our strains are sensitive, there are studies where the perceived resistance can reach 31.7% [76]. All strains were shown to have the *opr*D gene that encodes a substrate-specific outer membrane porin of *P. aeruginosa,* which allows the diffusion of basic amino acids, small peptides, and imipenem in the cell. A study by Estepaa et al. demonstrated that the high polymorphism observed in the *opr*D gene, as well as the presence of insertion elements truncating it, would inactivate the *opr*D porin, directly related to carbapenem resistance among strains [67]. Although all our strains have the *opr*D gene, there may be some polymorphism that interferes with the resistance to carbapenem mechanisms, which agrees with our results. To confirm this, there is the need to sequence the strains used and to perform a polymorphism analysis.

Regarding the presence of virulence factors, all strains have shown to have numerous virulence genes that are involved in different mechanisms (Figure 1). Several studies have investigated the presence of virulence genes in *P. aeruginosa* isolated from different sources, including urine samples. For example, one study found that the presence of certain virulence genes in *P. aeruginosa* isolated from urine samples was associated with increased mortality in patients with urinary tract infections [33,55,77]. The *exo*U, *exo*S, *exo*A, *exo*Y, and *exo*T are involved in the type III secretion system and encode different effectors that cause cytotoxicity at different levels. Only one isolate showed to have *exoU*, in contrast to the *exo*S (n = 25), *exo*A (n = 25), *exo*Y (n = 25), and *exo*T (*n* = 25) genes which have a high prevalence. The presence of *exo* genes is highly prevalent in *P. aeruginosa* strains once they are part of a type III secretion system, something intrinsic to this bacterium [51,56,62]. Studies reported a low to moderate prevalence of the *exo*U gene, while for our study it was low [78,79]. This gene does not appear in a constant way and is related to the type of infection. Interestingly, and accordingly to our results, the literature reports a low prevalence of this gene in urinary infections [80] and a high prevalence in other infections [81]. This contrast may be due to the differences in the studied infections (urinary tract infections vs. cystic fibrosis, for example) or personal characteristics (adults vs. children).

The *tox*A and *las*B genes were found in all strains, while *las*A genes did not appear in any strain. Usually, these genes encode proteins secreted by the type II secretion system, which plays a major role during infection. Our data agree with other works, where the prevalence of the *tox*A gene is high, around 100% in burn infections and 91.8% from isolates mainly from urine [57,82]. The prevalence of *las*B in these strains is also high. For the type I secretion system, the *apr*A gene appears in all strains and is responsible for the production of alkaline protease. Several studies have investigated the prevalence of LasB in *P. aeruginosa* strains isolated from different clinical and environmental sources. For example, a study analyzed 60 *P. aeruginosa* strains isolated from various sources, including clinical samples, water, and soil. The authors found that the presence of the *las*B gene was found in all strains [83]. Moreover, several LasB variants with different molecular weights and substrate specificities were identified, suggesting that *P. aeruginosa* has a diverse array of LasB enzymes that have evolved to adapt to different environments and hosts [84,85]. In addition to LasB, *P. aeruginosa* can secrete other proteases and lipases via various secretion systems, such as the type I, II, and III secretion systems. The type I secretion system is a one-step pathway that can transport proteins directly from the cytoplasm to the extracellular space. One of the key components of the type I secretion system is the *apr*A gene, which encodes a membrane-bound ATP-binding cassette (ABC) transporter that can export alkaline proteases and other virulence factors. A study by Fito-Boncompte and colleagues analyzed the distribution of type I secretion system genes in 52 *P. aeruginosa* strains isolated from cystic fibrosis patients. The authors found that the *apr*A gene was present in all strains, indicating that this gene is highly conserved and essential for *P. aeruginosa* virulence [86]. The *tssC* is among the genes that do not influence biofilm formation but are implicated in biofilm-specific antibiotic resistance [87]. As expected, it was found in all strains, since 92% of them are biofilm producers and have some resistance to antimicrobial agents.

All strains carry genes encoding alginate (*alg*D) and phospholipases C (*plc*H) that are normally involved in pulmonary infection and proinflammatory activities, respectively. In addition, the *alg*D was shown to have a wide dissemination, as reported by the literature [88]. Another highly prevalent virulence gene was *plc*H. In this study, it showed a prevalence of 100%, which is much higher compared to the other previous reports [61,82].

Only one strain was shown to have the *pil*B gene (4%), and some strains had the *pil*A gene (24%), which was expected since these genes have low dissemination, and their presence also varies with the type of infection [19,42,88]. Studies have shown that mutations in *pil*A or *pil*B can affect the ability of *P. aeruginosa* to form biofilms, adhere to surfaces, and cause infections. For example, several studies found that a mutant strain of *P. aeruginosa* with a deletion in the *pil*A gene was unable to form biofilms on abiotic surfaces and had reduced virulence in a mouse model of infection [89,90]. Similarly, a study found that a mutant strain of *P. aeruginosa* with a deletion in the *pil*B gene had reduced twitching motility and reduced biofilm formation [91]. Thereby, the genes *pil*A and *pil*B are important for the biosynthesis and assembly of T4P in *P. aeruginosa*, and mutations in these genes can affect the ability of the bacterium to form biofilms, adhere to surfaces, and cause infections.

Three distinct quorum sensing systems have been described in *P. aeruginosa:* two LuxIR homologs (*las* and *rhl*) and the 2-heptyl-3-hydroxy-4-(1H)- quinolone (PQS) [6]. In the present study, a high occurrence of QS genes was found. All strains presented multiple genes associated with the *las* system and *rhl* system. For the *las* system, the *las*R (n = 24) and *lasI* (n = 25), and for the *rhl* system, the *rhl*R (n = 23) and *rhl*I (n = 25) genes were found, in contrast to the *rhl*A/B gene that did not appear in any strain. Studies have shown that the Las and Rhl systems are interconnected and form a hierarchical regulatory network that controls the expression of downstream genes. The Las system is first activated during early stages of growth, while the Rhl system is activated later during the transition from exponential to stationary phase. In addition to their role in virulence and biofilm formation, the Las and Rhl systems are also involved in other physiological processes in *P. aeruginosa* [59,60]. For example, the Rhl system has been shown to regulate the expression of genes involved in iron acquisition and metabolism, and the Las system has been shown to regulate the expression of genes involved in stress response [92,93]. Realizing that these systems do not have any mutations in the genes helps us understand the normal development of *P. aeruginosa* under normal conditions [94]. The microtiter assay is the most used method for the analysis of biofilm biomass due to its accuracy and reproducibility [95,96,97]. Among the twenty-five isolates, 92% of the isolates produced biofilms, however 44% (*n* = 11) were low producers, 24% (*n* = 6) moderate producers, and 24% (*n* = 6) were high-producers (Figure 2). All data and calculations for obtaining the classification of biofilms can be found in Appendix A. Studies have shown that *P. aeruginosa* biofilm formation in human urine samples is associated with an increased risk of UTI [98,99,100]. *P. aeruginosa* biofilm formation in human urine samples is a significant issue in the treatment of UTIs and can lead to persistent infections and antibiotic resistance. Some studies used the crystal violet method to compare the ability of different *P. aeruginosa* strains to form biofilms. According to the studies’ findings, certain strains exhibited greater ability to form biofilms compared to others [101]. Another study used the crystal violet method to investigate the effect of different antibiotics on biofilm formation by *P. aeruginosa*. They found that some antibiotics were more effective than others at stopping biofilms [102]. Further research is needed to fully understand the mechanisms of *P. aeruginosa* biofilm formation in human urine samples and to develop more effective and targeted treatments for UTIs caused by this bacterium.

The motility of *P. aeruginosa* in human urine samples has been extensively studied, with much evidence indicating that its motility plays a critical role in its ability to cause infections. All isolates were assayed for motility. The swarming motility makes it possible to move across semi-solid surfaces in the presence of specific nutritional cues, and this movement appears to require motile flagella [42]. Some swarm colonies differed in morphology among clinical isolates. A study performed by Murray et al. showed that morphology was not related to bacterial swimming or twitching phenotypes in any consistent fashion [42]. Within our series of clinical isolates, swimming was the movement that reached the greatest distances in contrast to the other movements (Figure 3).

To better characterize the different movements performed by each isolate, a single dots analysis was performed. Each isolate demonstrates that they all have movement but that they are more conducive to one type than the rest. It is clear that all isolates perform better when doing the swim movement than they do while swarming or twitching, as shown in the Figure 4. Isolate 7 had a better performance in the twitch movement. Isolate 9 performed better in swarm, and isolates 15 and 22 had equal performance in swarm and swim.

*P. aeruginosa* is a highly motile bacterium that exhibits a variety of motility mechanisms. The most common and well-studied motility mechanism in *P. aeruginosa* is called twitching motility, which involves the extension and retraction of type IV pili on the cell surface. Twitching motility allows *P. aeruginosa* to crawl along surfaces, including solid surfaces and the surfaces of host cells [103]. Other motility mechanisms in *P. aeruginosa* include swimming motility and swarming motility; however, compared to twitching motility, these mechanisms are less common in this bacterium [104]. However, in our study, swimming was the one that presented the best performance. *P. aeruginosa* is known to exhibit both types of motilities; however, its swimming motility is sometimes more noticeable than its twitching motility for several reasons. Firstly, *P. aeruginosa* possesses a flagellum, a whip-shaped structure that facilitates its movement in liquid environments. The flagellum operates in a rotational motion, propelling the bacterium in a forward direction, and its movement is more noticeable than its twitching motility. [104]. Secondly, swimming motility is faster than twitching motility, with *P. aeruginosa* being able to move several body lengths per second in liquid environments. Twitching motility, on the other hand, is slower and more limited in range [105]. Finally, swimming motility enables the bacterium to move quickly and efficiently through mucus and other bodily fluids, allowing it to colonize and infect different parts of the body. In contrast, twitching motility is thought to be more important for surface attachment and biofilm formation, which are less critical for pathogenesis in the case of *P. aeruginosa* [106].

Other studies have shown that the motility of *P. aeruginosa* present in urine can be influenced by factors such as the presence of other bacteria, the presence of antimicrobial agents, and the pH of the urine. These findings indicate that the movement *of P. aeruginosa* in human urine samples is a multifaceted and ever-changing process, influenced by various factors. It is crucial to comprehend the bacterium’s motility to establish effective strategies to limit its spread and prevent infections [36,107,108]. This could justify our results since, under normal conditions, *P. aeruginosa* adopts a swimming movement as it allows the bacteria to migrate to new locations in a state of infection and to penetrate the host tissues. On the contrary, swarm and twitch movements are often used in order to overcome environmental factors, such as the presence of nutrients, pH, and the availability of iron.

In *P. aeruginosa*, the transcription of genes encoding extracellular proteases LasB (*las*B) and LasA (*las*A), alkaline protease (*apr*A), and protease IV is upregulated in the swarm [109]. In a study where the relationship of casein hydrolytic activity and bacterial swarm-secreted protease activity is evaluated, isolates that exhibited large swarm diameters exhibited increased secreted caseinolytic activity relative to isolates that exhibited reduced or absent swarm (*p* < 0.0001). Since protease secretion was tested under non-swarming conditions, this result indicated that protease secretion is inherently higher in swarming-capable strains and not simply upregulated during swarming [42]. Furthermore, no mutants for the single protease genes were defective for swarming. In our study, since the genes that are related to caseinolytic activity in the swarm movement were detected, there was possibly a greater secretion of extracellular proteases, alkaline protease and protease IV during bacterial growth than in the other movements.

In general, measuring the movement diameter of *P. aeruginosa* can provide valuable information about its motility and behavior, which can have important implications for understanding its pathogenicity and developing treatments for infections caused by this bacterium. For example, several studies were executed regarding the motility of *P. aeruginosa* in the presence of an electric field using microfluidic devices. The authors found that bacteria were attracted to the anode and repelled by the cathode, and that the movement diameter of *P. aeruginosa* was influenced by the strength of the electric field [110]. On the other hand, studies that utilized microscopy to observe the motility of *P. aeruginosa* found that its walking pattern depended on the motility of its type IV pili. The authors reported that the walking diameter of *P. aeruginosa* varied depending on the strain and the surface properties of the bacterial substrate [111,112,113,114]. Understanding the basic movements of these strains will make it possible in future studies to investigate the molecular mechanisms underlying motility in *P. aeruginosa*, as well as environmental adaptation.

The reproducibility evaluation test using the generalized Bland–Altman plot is based on assessing the agreement between quantitative measurements by plotting the difference between them against their average (Figure 5). It is a statistical method commonly used to evaluate the level of agreement between two sets of measurements and to identify any systematic bias or random error between them. The plot provides a visual representation of the level of agreement between the measurements, and the analysis can help determine if the two methods are interchangeable. The reproducibility of the motility tests showed a media average coefficient is 0.199, 0.966 and 0.118, for the swarm, swim, and twitch test, respectively. We can infer that we do not have a proportion bias, meaning that the values are evenly distributed between the media average.

## 4. Conclusions

This study focused on the isolation and characterization of *Pseudomonas aeruginosa* strains from human urine infections. Among the twenty-five isolates analyzed, 16% were found to be multidrug-resistant. The prevalence of resistance to carbapenem antibiotics was low, in line with previous reports. However, the presence of resistance genes such as *bla*_NDM_, *bla*_SPM_, bla_VIM-VIM2_, and *bla*_KPC_ indicates the potential for the emergence and spread of these resistant strains.

Virulence gene analysis revealed the presence of several genes associated with different virulence mechanisms. The *exo* genes, which are involved in the type III secretion system, showed a high prevalence, except for the *exo*U gene, which was present in only one isolate. The *tox*A and *las*B genes were found in all strains, while the *las*A gene was absent. Other virulence genes such as *apr*A, *tss*C, *alg*D, *plc*H, *pil*A, and *pil*B exhibited varying prevalence rates.

Generally, the findings of this study provide valuable insights into the antimicrobial resistance and virulence characteristics of *P. aeruginosa* strains isolated from urine infections. The presence of multidrug-resistant strains and the high prevalence of certain resistance and virulence genes underscore the importance of surveillance and infection control measures to prevent the spread of these problematic bacteria. Further studies and genetic analyses are needed to elucidate the relationships between the isolates and to better understand the mechanisms underlying their resistance and virulence phenotypes.

## Figures and Tables

**Figure 1 microorganisms-11-01345-f001:**
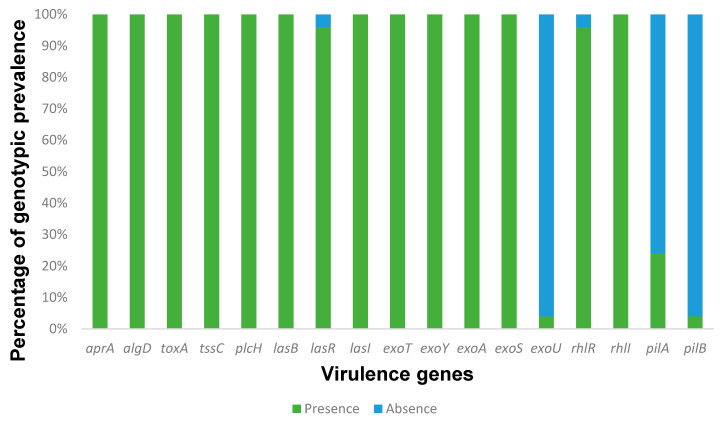
The percentage of each virulence gene found within the *P. aeruginosa* isolates derived from UTIs at the Medical Centre of Trás-os-Montes and Alto Douro, Vila Real, Portugal, between September 2021 and June 2022. Results revealed that all isolates contained multiple genes associated with various virulence mechanisms. However, some genes such as lasR, exoU, rhIR, pilA, and pilB were not detected in all isolates, particularly exoU and pilB, which exhibited the lowest prevalence among the isolates.

**Figure 2 microorganisms-11-01345-f002:**
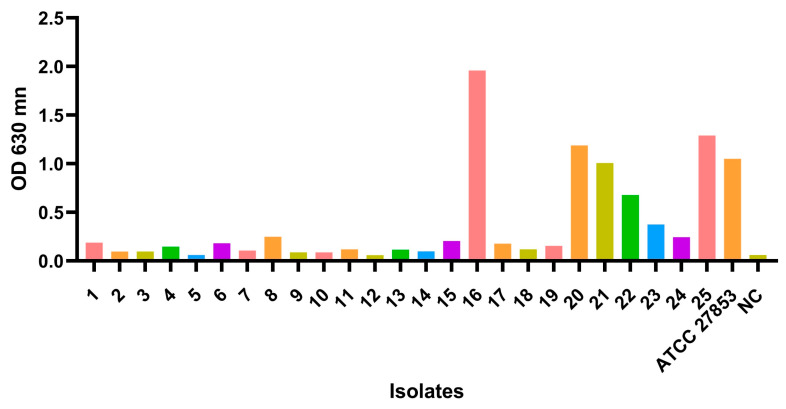
Measurement of biofilm formation in urinary tract infection strains. Biofilm formation was measured using the crystal violet method as described under Materials and Methods. The control Pseudomonas aeruginosa ATCC^®^ 27,853 was used as positive control in the formation of biofilms and TSB as a negative control (NC).

**Figure 3 microorganisms-11-01345-f003:**
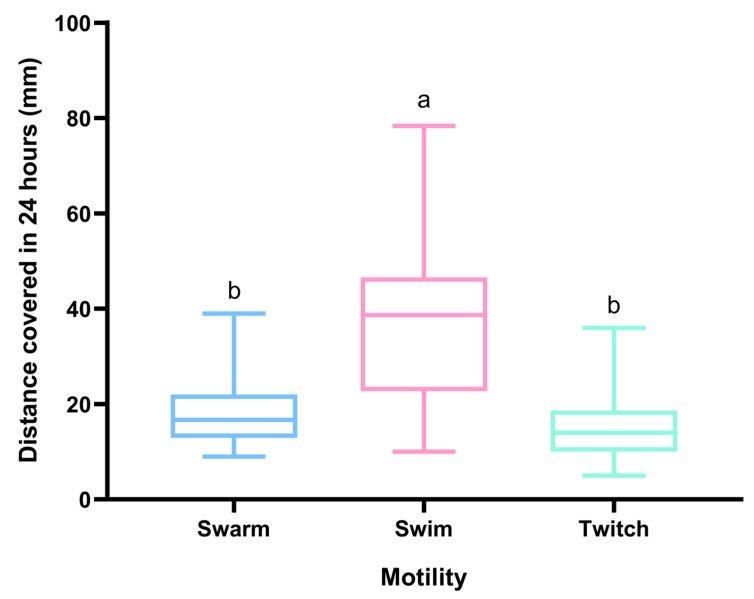
Average zone of growth (diameter mm) was measured for all isolates after 24-h incubation. Statistical significance was determined using Tukey’s multiple comparison test (*p* < 0.0001). Two statistically significantly different groups have emerged, where a > b.

**Figure 4 microorganisms-11-01345-f004:**
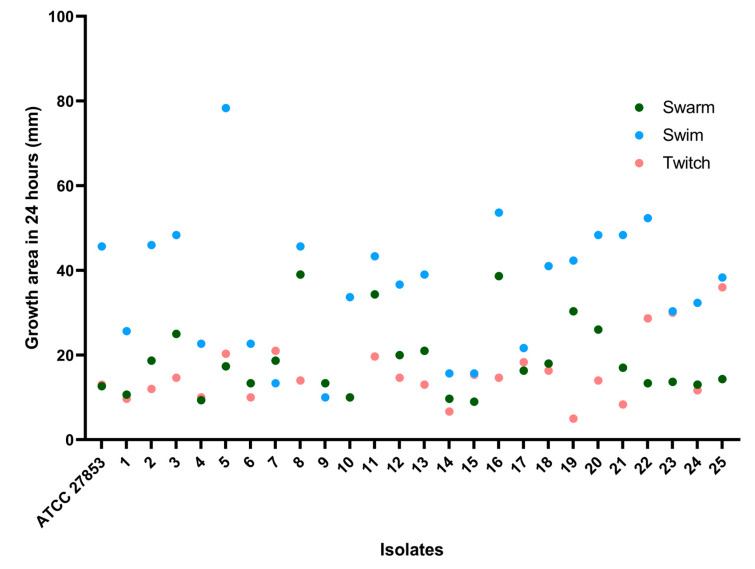
The zones of motility measured as diameter determined after 24 h on agar plates for twitching, swarming, and swimming for each isolate.

**Figure 5 microorganisms-11-01345-f005:**
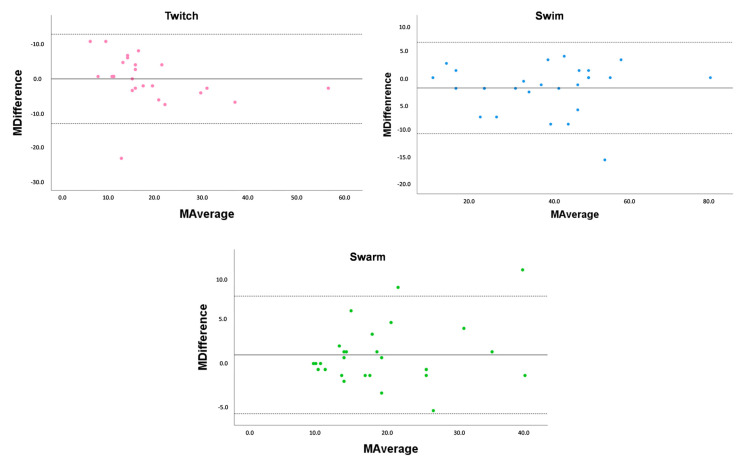
Reproducibility assessment of twitch, swim, and swarm motility using the generalized Bland–Altman Plot. Boundaries for this plot are ISO-defined reproducibility limits, defined as ±1.96 SD. Each isolate tested in triplicate.

**Table 1 microorganisms-11-01345-t001:** Nucleotide sequence of primers for PCR reaction and size of amplified DNA fragment for antibiotic resistance and virulence genes.

Name	Sequence (5′ → 3′)	Length (bp)	Reference
*bla* _TEM_	F: ATTCTTGAAGACGAAAGGGC R: ACGCTCAGTGGAACGAAAAC	1150	[45]
*bla* _SHV_	F: CACTCAAGGATGTATTGTG R: TTAGCGTTGCCAGTGCTCG	885	[46]
*bla* _CTX_	F: CGATGTGCAGTACCAGTAA R: TTAGTGACCAGAATCAGCGG	585	[47]
*bla* _PER_	F: ATGAATGTCATTATAAAAGC R: AATTTGGGCTTAGGGCAGAA	920	[48]
*bla* _SME_	F: ACTTTGATGGGAGGATTGGC R: ACGAATTCGAGCATCACCAG	551	[49]
*bla* _KPC_	F: GTATCGCCGTCTAGTTCTGC R: GGTCGTGTTTCCCTTTAGCC	638	[50]
*bla* _Smp_	F: AAAATCTGGGTACGCAAACG R: ACATTATCCGCTGGAACAGG	271	[51]
*bla* _Vim_	F: TTTGGTCGCATATCGCAACG R: CCATTCAGCCAGATCGGCAT	500	[52]
*bla* _Vim-2_	F: AAAGTTATGCCGCACTCACC R: TGCAACTTCATGTTATGCCG	815	[53]
*bla* _NDM_	F: GGTTTGGCGATCTGGTTTTC R: CGGAATGGCTCATCACGATC	621	[50]
*bla* _OXA_	F: CCAAAGACGTGGATG R: GTTAAATTCGACCCCAAGTT	813	[46]
*aac*(6′)-Ie-*aph*(2″)-Ia	F: CCAAGAGCAATAAGGGCATA R: CACTATCATAACCACTACCG	220	[25]
*aph*(3′)-IIIa	F: GCCGATGTGGATTGCGAAAA R: GCTTGATCCCCAGTAAGTCA	292	[25]
aac(3)-II	F: ACTGTGATGGGATACGCGTC R: CTCCGTCAGCGTTTCAGCTA	237	[24]
aac(3)-III	F: CACAAGAACGTGGTCCGCTA R: AACAGGTAAGCATCCGCATC	195	[24]
aac(3)-IV	F: CTTCAGGATGGCAAGTTGGT R: TACTCTCGTTCTCCGCTCAT	286	[25]
*ant*(4′)-Ia	F: GCAAGGACCGACAACATTTC R: TGGCACAGATGGTCATAACC	165	[25]
*ant*(2′)-I	F: ATGTTACGCAGCAGGGCAGTCG R: CGTCAGATCAATATCATCGTGC	188	[24]
* oprD *	F: TCCGCAGGTAGCACTCAGTTC R: AAGCCGGATTCATAGGTGGTG	191	[54]
*pil*B	F: TCGAACTGATGATCGTGG R: CTTTCGGAGTGAACATCG	408	[55]
*pil*A	F: ACAGCATCCAACTGAGCG R: TTGACTTCCTCCAGGCTG	1675	[55]
*apr*A	F: ACCCTGTCCTATTCGTTCC R: GATTGCAGCGACAACTTGG	140	[56]
*tox*A	F: GGTAACCACGTCAGCCACAT R: TGATGTCCAGGTCATGCTTC	352	[57]
*tss*C	F: CTCCAACGACGCGATCAAGT R: TCGGTGTTGTTGACCAGGTA	150	[58]
*plc*H	F: GCACGTGGTCATCCTGATGC R: TCCGTAGGCGTCGACGTAC	608	[55]
*las*A	F: GCAGCACAAAAGATCCC R: GAAATGCAGGTGCGGTC	1075	[55]
*las*B	F: GGAATGAACGAAGCGTTCTC R: GGTCCAGTAGTAGCGGTTGG	284	[59]
*las*R	F: CGGGTATCGTACTAGGTGCATCA R: GACGGGAAAGCCAGGAAACTT	1100	[56]
*lasI*	F: ATGATCGTACAAATTGGTCGGC R: GTCATGAAACCGCCAGTCG	605	[60]
*exo*U	F: ATGCATATCCAATCGTTG R: TCATGTGAACTCCTTATT	2000	[56]
*exo*S	F: CGTCGTGTTCAAGCAGATGGTGCTG R: CCGAACCGCTTCACCAGGC	444	[61]
*exo*A	F: GACAACGCCCTCAGCATCACCAGC R: CGCTGGCCCATTCGCTCCAGCGCT	396	[62]
*exo*Y	F: CGGATTCTATGGCAGGGAGG R: GCCCTTGATGCACTCGACCA	289	[62]
*exo*T	F: AATCGCCGTCCAACTGCATGCG R: TGTTCGCCGAGGTACTGCTC	159	[62]
*rhl*R	F: CAATGAGGAATGACGGAGGC R: GCTTCAGATGAGGCCCAGC	730	[60]
*rhl*I	F: CTTGGTCATGATCGAATTGCTC R: ACGGCTGACGACCTCACAC	625	[60]
*rhl*A/B	F: TCATGGAATTGTCACAACCGC R: ATACGGCAAAATCATGGCAAC	151	[63]
*alg*D	F: CGTCTGCCGCGAGATCGGCT R: GACCTCGACGGTCTTGCGGA	313	[55]

**Table 2 microorganisms-11-01345-t002:** Characteristics of the *Pseudomonas aeruginosa* isolates derived from urine samples.

Isolate	Antimicrobial		
	Susceptible	Resistant	Genotype
1	TOB, AK, CN, CS	PTZ, CAZ, ATM, CIP, TTC	*bla*_NDM_, *bla*_OXA_, *opr*D
2	TOB, AK, CN, CS		*bla*_OXA_, *opr*D
3	TOB, AK, CS	CN	*bla*_OXA_, *aph*(3′)-IIIa, *opr*D
4	TOB, AK, CN, CS	LEV	*bla*_NDM_, *opr*D
5	TOB, AK, CN, CS		*bla*_CTX-M-UN_, *opr*D
6	TOB, AK, CN, CS		*bla*_KPC_, *bla*_SPM_, *opr*D
7	TOB, AK, CN, CS		*bla*_KPC_, *opr*D
8	TOB, AK, CN, CS	PTZ, CAZ	*bla*_KPC_, *bla*_SPM_, *opr*D
9	TOB, AK, CN, CS		*bla*_KPC_, *bla*_NDM_, *bla*_SPM_, *opr*D
10	TOB, AK, CN, CS	PTZ, CAZ, ATM, TTC	*bla*_KPC_, *bla*_SPM_, *bla*_VIM2_, *opr*D
11	TOB, AK, CS	CN	*bla*_KPC_*, bla*_SPM_, *bla*_OXA_, *opr*D
12	TOB, AK, CN, CS		*bla*_CTX-M-UN_, *bla*_KPC_, *opr*D
13	TOB, AK, CN, CS		*bla*_KPC_, *opr*D
14	TOB, AK, CN, CS	PTZ, CAZ, CIP	*bla*_KPC_, *opr*D
15	TOB, AK, CN, CS		*bla*_KPC_, *bla*_SPM_, *opr*D
16	TOB, AK, CN, CS		*bla*_SPM_, *bla*_VIM_, *opr*D
17	TOB, AK, CS	CN	*bla*_KPC_, *bla*_NDM_, *bla*_SPM_, *opr*D
18	TOB, AK, CN, CS		*bla*_SPM_, *bla*_VIM2_, *opr*D
19	TOB, AK, CN, CS		*bla*_KPC_, *opr*D
20	TOB, AK, CN, CS		*bla*_KPC,_*bla*_SPM_, *opr*D
21	TOB, AK, CN, CS	IMI	*bla*_SPM_, *opr*D
22	TOB, AK, CN, CS		*bla*_SPM_, *opr*D
23	TOB, AK, CN, CS		*bla*_SPM_, *opr*D
24	TOB, AK, CS	PTZ, CN, TTC, PRL	*bla*_SPM_, *opr*D
25	TOB, AK, CN, CS		*bla*_SPM_, *opr*D

Ceftazidime (CAZ), cefepime (FEP), amikacin (AK), gentamicin (CN), tobramycin (TOB), doripenem (DOR), imipenem (IMI), meropenem (MEM), aztreonam (ATM), ciprofloxacin (CIP) piperacillin (PRL), colistin (CS), levofloxacin (LEV) and ticarcillin–clavulanic acid (TTC).

## Data Availability

Not applicable.

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
