# Peer review of "Study of Antimicrobial Resistance, Biofilm Formation, and Motility of *Pseudomonas aeruginosa* Derived from Urine Samples"

_microorganisms, 2023, doi:10.3390/microorganisms11051345_

Round 1

Reviewer 1 Report (New Reviewer)

The manuscript entitled « Study of Antimicrobial Resistance, Biofilm formation and mo tility of Pseudomonas aeruginosa derived from urine samples» is devoted to microbial biofouling by Pseudomonas aeruginosa of indwelling catheters.

 Pseudomonas aeruginosa is a major human pathogen  and has extensive resistance to antibiotics, for this reason, this study is important and relevant.

The manuscript seemed interesting, and in general it corresponds to the aims and scopes of the Microorganisms journal.

I have a few remarks and comments on the manuscript.

1. The abstract should be rewritten because it looks very general and does not reflect the results obtained by the authors.

2. The text looks generally unfinished, the authors should pay attention to its formatting. The text contains edits made by the authors that make it difficult to read the manuscript.

3. The introduction is generally well written, but should be made more logical. At the beginning, more general information should be given, from which the relevance of the work follows, and then a description of the biofilm formation systems and a description of the genome features responsible for the processes in biofilms should be given.

4. L 187-193 in the methodology look incomprehensible, the authors should describe the Biofilm-formation abilty method in more detail and clearly

5. The Swimming Motility methodology is also not very clear.

6. and motility at this interface was subsequently measured [37] by what method?

7. Why was Klebsiella pneumoniae used as a negative control?

8. Authors should provide a table of isolated strains, describe how similar they are

9. I advise you to reorganize the table 2. How important is the month of isolation? Are the data in Table 2 based on laboratory tests, or only on the basis of antibiotic resistance genes found?

10. Fig1 aprA etc are the names of the genes, but where is the name of the isolates? How was the presence / absence considered. The picture is not very clear.

11. Is Figure 3 an average of all isolates? In my opinion, it is not very informative and can be transferred to supplementary

12. Is Figure 5 an average of all isolates? A more detailed understanding of the figure and more understandable conclusions from it are required.

13. Conclusions should be rewritten. The authors worked with 25 isolates, the conclusion should be written which isolates are the most dangerous, in terms of antibiotic resistance and biofilm formation. The authors received a lot of valuable data, but did not properly comprehend them. I would advise you to single out a separate chapter of the discussion in order to compare and understand the data obtained. Are there strains in isolated strains that can cause superinfection, with maximum resistance to antibiotics and forming the strongest biofouling?

English is generally good

Author Response

The authors thank all reviewers for their constructive comments, which allowed significant improvement of the manuscript. We proceeded with the revision of the manuscript in the light of the comments received and brief responses to the reviewers’ comments are included. All the modifications in the text are marked using “Track Changes” function. 

Reviewer 2 Report (New Reviewer)

Dear authors,

Your paper brings together new essential results regarding antibiotic-resistance profiles in P. aeruginosa strains recovered from urine samples. This is a very interesting manuscript. However, I have some minor recommendations:

1. Line 88 -> I suggest to modify here: " .. the colonization of abiotic and biotic surfaces".

2. Line 118 -> Please verify in the text the name of the bacteria that need to be in italics.

3. Verify the abbreviation in the text, to ensure that only the first time when is mentioned a term appear in the full form.

4. Please add the full name of this abbreviation: CHTMAD -line 534.

5. Please verify the reference list. For documents co-authored by a large number of persons (more than 10 authors), you can either cite all authors, or cite the first ten authors, then add a semicolon and add ‘et al.’ at the end.

The paper requires only minor English editing.

Author Response

The authors thank all reviewers for their constructive comments, which allowed significant improvement of the manuscript. We proceeded with the revision of the manuscript in the light of the comments received and brief responses to the reviewers’ comments are included. All the modifications in the text are marked using “Track Changes” function. 

Round 2

Reviewer 1 Report (New Reviewer)

in the manuscript corrected by the author, my main remarks were eliminated, so I propose to accept it in this form.

This manuscript is a resubmission of an earlier submission. The following is a list of the peer review reports and author responses from that submission.

Round 1

Reviewer 1 Report

The Authors selectively corrected the manuscript according to some of the remarks, surprisingly neglecting (in purpose?) some crucial things that still has to be corrected, including necessary information on controls. Why is that? Haven't you used any? Moreover, they introduced new issues, providing even worse version of the manuscript now and making the whole review process quite disappointing. See below:

Line 38 - "All the 92%" what does that mean???

Line 150 "were isolated derived", either "isolated/derived from urinary tract infections" or "isolated/derived from urine samples" not both,

Line 197 - number 5, again.

Line 226 - what is the range of DNA concentration that you had measured?, again.

Information on tssC gene is missing, again.

Sentences/paragraphs should not start with numbers (never), again.

Italics missing in a number of sites, typos, missing comas, etc.

What does "full susceptible isolates" mean?

Caption of Figure 1 sounds awkward.

The same with current caption of Figure 2 and current figure 4.

Lines 275-277 - any reference for this statement, which is actually wrong, again.

What about the figure illustrating AST results, again.

What was/were the control strains/DNA for the virulence genes and antimicrobial resistance genes detection?, again. If there was none, the credibility of the results is controversial and the paper should be rejected.

Figure 3, y axis legend "covered" sounds awkward.

I suggest placing the negative results of the genes investigation along with positives, again.

Line 353 - Reference [80] does not provide information on LasB synthesis, but on on lasB gene presence!

Line 412 - the phrase "proficient at forming biofilms" is a personification of bacteria.

Line 429 - this sentence/explanation is unclear, therefore confusing and misleading, in my opinion, it needs to be rephrased/corrected, again,

Line 459 and 461 – the phrases “motility of P. aeruginosa in urine”sound awkward, again,

Discussion: the mentioning of information on gene expression investigation (in the future) would be useful, again.

Conclusion section is definitely too long, I suggest to replace some of the sentences into Discussion (especially limitations of the study) section or delete, again,

"One explanation is that the swimming motility and twitching motility are two different types of bacterial movement" it is obvious, making the sentence sounds awkward.

"Finally, P. aeruginosa is an opportunistic pathogen that can cause infections in humans." what is the reason of placing this sentence here?

Author Response

Dear Reviewer,

I am writing to inform you that we have taken into consideration all the suggestions proposed in your review and have made the necessary corrections to our manuscript. We appreciate your valuable feedback, and we believe that it has helped to improve the quality of our work significantly.

We have carefully reviewed each of your comments and have made the required changes accordingly. We have also addressed all the concerns that you have raised in your review. We are confident that our manuscript now meets the high standards expected for publication in your esteemed journal.

Once again, we thank you for your insightful review and appreciate the time and effort you have put into providing us with constructive feedback. We hope that our revised manuscript meets your expectations.

Sincerely,

Patricia Poeta

Reviewer 2 Report

Lines 33-34 Should describe where the isolates came from and how many were tested.

I think 38-39 needs to be revised. There still is not a negative control on the biofilm figure but most look like background, therefore, I do not think 92% were capable of biofilm formation. How was a positive determined? If statistics were used based on positive and negative controls they should be described.

Figure legend 1 is still problematic. It should read something similar to:

The percentage of each virulence gene found within the P. aeruginosa isolates derived from UTIs at the Medical Centre of Trás-os-Montes and Alto Douro, Vila Real, 151 Portugal between September 2021 and June 2022. Why was this information not presented on per-isolate bases like in table 1?

Absence is still spelled wrong on Figure 1.

The figure legends still lack important information.

Another example is figure 4 which should read something like: The zones of motility measured as diameter determined after 24 hours on agar plates for twitching, swarming, and swimming for each isolate. I am not sure why these dots are connected. isolate 1 is independent of isolate 2 etc. I would remove these connections.

This question was not addressed in the previous review. 

Do isolates that have those genes pilA and pilB have significantly better twitching phenotypes than those that do not? Does the control P. aerugionsa have all genes to twitch?

All other concerns were addressed in the first review.

Author Response

(The authors gave the same response as above.)
